# Hopes and Limits of Adipose-Derived Stem Cells (ADSCs) and Mesenchymal Stem Cells (MSCs) in Wound Healing

**DOI:** 10.3390/ijms21041306

**Published:** 2020-02-14

**Authors:** Loubna Mazini, Luc Rochette, Brahim Admou, Said Amal, Gabriel Malka

**Affiliations:** 1Laboratoire Cellules Souches et Régénération Cellulaire et Tissulaire, Centre interface Applications Médicales (CIAM), Université Mohammed VI Polytechnique, Ben-Guerir 43 150, Morocco; gabriel.malka@um6p.ma; 2Equipe d’Accueil (EA 7460), Physiopathologie et Epidémiologie Cérébro-Cardiovasculaires (PEC2), Faculté des Sciences de Santé Université de Bourgogne—Franche Comté, 7 Bd Jeanne d’Arc, 21000 Dijon, France; luc.rochette@u-bourgogne.fr; 3Laboratoire d’immunologie, Centre de Recherche Clinique, Faculté de Médecine et Pharmacie, Université Cadi Ayyad, Centre Hospitalier Universitaire, Marrakech 40 000, Morocco; admou.fmpm@gmail.com; 4Service de dermatologie, Faculté de Médecine et Pharmacie, Université Cadi Ayyad, Centre hospitalier universitaire, Marrakech 40000, Morocco; saidamalb@hotmail.com

**Keywords:** adipose derived stem cells, skin, regeneration, differentiation, wound healing, aging, rejuvenation, microenvironment

## Abstract

Adipose tissue derived stem cells (ADSCs) are mesenchymal stem cells identified within subcutaneous tissue at the base of the hair follicle (dermal papilla cells), in the dermal sheets (dermal sheet cells), in interfollicular dermis, and in the hypodermis tissue. These cells are expected to play a major role in regulating skin regeneration and aging-associated morphologic disgraces and structural deficits. ADSCs are known to proliferate and differentiate into skin cells to repair damaged or dead cells, but also act by an autocrine and paracrine pathway to activate cell regeneration and the healing process. During wound healing, ADSCs have a great ability in migration to be recruited rapidly into wounded sites added to their differentiation towards dermal fibroblasts (DF), endothelial cells, and keratinocytes. Additionally, ADSCs and DFs are the major sources of the extracellular matrix (ECM) proteins involved in maintaining skin structure and function. Their interactions with skin cells are involved in regulating skin homeostasis and during healing. The evidence suggests that their secretomes ensure: (i) The change in macrophages inflammatory phenotype implicated in the inflammatory phase, (ii) the formation of new blood vessels, thus promoting angiogenesis by increasing endothelial cell differentiation and cell migration, and (iii) the formation of granulation tissues, skin cells, and ECM production, whereby proliferation and remodeling phases occur. These characteristics would be beneficial to therapeutic strategies in wound healing and skin aging and have driven more insights in many clinical investigations. Additionally, it was recently presented as the tool key in the new free-cell therapy in regenerative medicine. Nevertheless, ADSCs fulfill the general accepted criteria for cell-based therapies, but still need further investigations into their efficiency, taking into consideration the host-environment and patient-associated factors.

## 1. Introduction

Multipotent mesenchymal/stromal stem cells (MSCs) have been identified as residual stem cells in almost all adult organs, especially within adipose tissue (AT). These cells present, in vitro, the typical mesenchymal cell characteristics and are isolated within the stromal vascular fraction (SVF) [1,2]. Mainly called adipose derived stem cells (ASCs or ADSCs) and isolated in a less invasive and more reproducible manner, these cells are more proliferative and have immunosuppressive properties that are able to inactivate T cells [3,4]. ADSCs were demonstrated to differentiate into the adipogenic lineage when compared to bone marrow (BM)- and umbilical cord (UC)-MSCs, however their multipotency is actually more appreciated for ectodermic and endodermic tissue repair [4,5,6].

As evidenced by most reports, ADSCs are able to secrete a rich secretome, whereby cell proliferation and differentiation, migration, and an improvement to the cellular and microenvironment protection occurred [7,8,9,10,11,12,13]. This secretome corresponds to a panel of trophic factors, such as cytokines, growth factors, and chemokines, which allow ADSCs to act as paracrine tools that are more likely than cell replacement. Used as exosomes or conditioned-media, this secretome has opened the way to a newly emerged, cell-free therapy [13,14].

Recently, ADSCs were identified within subcutaneous tissue [15]. Their presence allows us to expect them to play a pivotal role in skin repair and regeneration. Indeed, there was evidence for the critical role of ADSCs in maintaining the structure of skin tissue, even as a physiological response to local injury or as rejuvenating mechanisms by seeding younger cells to the outer of the epidermis [5,15,16,17]. Identified within the basal layer where they self-renewed and differentiated to continuously settle the epidermis with keratinocytes, fibroblasts, and melanocytes [18,19], these cells might influence the physiological characteristics of the injured skin and presented with a great ability in migration and were recruited into wounded sites [11,20,21,22]. ADSCs have been shown to differentiate into keratinocytes, dermal fibroblasts (DF), and other skin components [15,23,24].

Additionally, ADSCs might be influenced in their ability to regenerate the injured tissue. In skin aging, these cells are expected to reduce their proliferation while their differentiation ability remains conserved, with a decrease of ECM secretion and an increase of cell apoptosis and accumulation of senescent cells [25,26]. Senescent cells secrete a specific senescent secretome [27], resulting in an increase in aging-associated cell symptoms that are morphologically apparent by the loss of skin elasticity, thickness, and increasing wrinkles [28]. Moreover, aging also impacts other epithelial cells that reduce their replicative capacity and induce reactive oxygen species (ROS) accumulation, as well as decreasing DF size and function [29,30,31].

Finally, the changes in the cell composition of the dermis and the ability of different epithelial cells to secrete specific growth factors such as TGF-β, GDF11, GDF15, b-FGF, VEGF, MMP-1, MMP-2, MMP-9, and extracellular matrix (ECM) proteins confer the possibility of establishing a balance between cell regeneration and cell rejuvenation to the ADSC’s microenvironment. In this review, we attempt to emphasize the mutual interactions between ADSCs, their surrounding cells, ECM proteins, and the panel of the microenvironment growth factors, as well as to determine their role in the regulation and the induction of cell regeneration in cases of injury and aging. Controlling this microenvironment might raise a potential to increase cell functionality and life span in order to counterbalance the physiological symptoms related to aging-associated diseases. This might open the way to a new era of managing the organ life span for promising therapeutic advancements.

## 2. The Skin between the Theory and the Physiology of Aging

Skin morphology is the illustration of observable time passing by epidermal atrophy related to wrinkles appearance, reduction of dermal thickness, and pigmentation defects. This extrinsic aging also reflects an intrinsic aging associated with structural and proliferative deficiencies. The decrease in the number of epithelial cells, including DF, Langerhans cells, and melanocytes, together with ECM degradation, results in an impairment of skin integrity and youth. Mutual interactions between DF and epidermal cells are closely related to skin integrity and are indicators of cellular and molecular changes during aging [32].

Epidermal cells and DF play a critical role in defining skin architecture and function. ECM proteins, mainly produced by DF and ADSCs, are composed of glycosaminoglycans, collagen type I, type III, and elastin, and are continuously modified by physiological and extrinsic factors. Collagen production is also decreased simultaneously with its degradation, leading to quantitative and qualitative changes in collagen fibers, and thus impacting the dermis structure [25,33]. Collagen I and III are degraded by the matrix metalloprotease 1 (MMP-1), the most ubiquitous endopeptidase secreted by keratinocyte, DF, and endothelial cells. However, the degradation of elastin fibers is the other representation of the dermis atrophy and it is connected to photoaging and is induced by MMP-2, -3, -9, and -12 [34,35]. These MMP levels increase with age and are modulated by MMP (TIMPs) tissue inhibitors. Their ratio is balanced during aging, in favor of MMP accelerating collagen deficiencies [36].

External physiological factors or UV are responsible for the degradation of ECM leading to an increase in enzymatic activity associated with collagen degeneration and loss of mechanical functions such as elasticity [37]. The skin might also be physiologically predisposed to accelerated aging and carcinogenesis; this is the case for various genetic syndromes that favored DNA damage or telomere dysfunction and cellular senescence. The decline in DNA repair ability, the increase of oxidative stress, and the shortening of the telomeres, may drive cells towards senescence [38].

Other intrinsic factors are known to impair physiological functions of the skin and are associated with cell senescence, including DNA damage [39,40], telomeres shortening [41], and reactive oxygen species (ROS) production [42]. All these processes show major roles in inducing tissue-aging and carcinogenesis [43,44].

## 3. Role of the Microenvironment in ADSCs Induction

ADSCs reside in vivo within adipose tissue in a specific location called “stem cell niche” where they are closely built-in with the ECM and the other supporting cells [45]. The relationship of ADSCs with this surrounding microenvironment and the mechanisms of action involved still requires further elucidation. In wound healing or aging, this microenvironment conditioned and modulated the involvement of ADSCs in the different mechanisms associated with skin tissue repair and anti-aging.

The microenvironment may modulate the biological properties and the ability of ADSCs to proliferate, differentiate, and migrate to restore cellular age defects and repair wound healing [46,47,48]. In fibrin 3D skin substitutes, the integration of ADSCs into the hypodermis leads them to differentiate into adipocytes, thus impacting the cell behavior of the upper epidermis layer of the substitute in vitro [49]. Many investigations went even further, using precursors of ECM, biomaterials scaffold, hydrogels, and electrospinning procedures to induce rapid wound closure and to increase epidermis thickness. However, an interesting approach has been presented by Chan et al., who used different biomaterials in one skin substitute to drive ADSC differentiation into different cell lineages [50].

In injured sites, ADSCs were recruited and were migrated by increasing their expression of CXCR-4 molecules, which led to modulating the inflammatory phenotype of local immune cells to a healing anti-inflammatory one. Their presence was critical to drive the inflammatory profile of macrophages, T, B, and dendritic cells initiating, and thus the proliferation and remodeling phases of wound healing [51,52]. Their action was likely activated by TGF-β or GDF11 or both of them [53]. Recently, ADSCs have been transfected with MicroRNA(miR)-146a and the resulting miR-enriched secretome strongly presented with angiogenic and anti-inflammatory capacities [54].

Additionally, preconditioning with inflammatory or pro-inflammatory cytokines has improved their responses to cancer and inflammation in addition to increasing their survival [55,56,57,58]. Several reports are in favor of optimizing the therapeutic benefit of ADSCs because of cytokine combination, including TGF-β and TNF-α. TGF-β was targeted by the integrin αβ6 and was secreted by the epithelial cells to modulate skin immune surveillance and the ADSC microenvironment [59]. The latter was dependent of IL-6 when initiating the inflammatory phase of wound healing. On this fact, ADSCs were likely stimulated by IL-6 to autoinduce and their secretion of TNF-α, b-FGF, VEGF, TLR2, TLR4, IL-6, TGF-β, and GDF11.

DF and ADSCs were demonstrated to modulate and address their microenvironment by secreting the growth factor as TGF-β and ECM to optimize keratinocyte and DF involvement in the healing process [46,47,48,60]. However, ADSCs also interacted with microvascular endothelial cells to secrete the increasing levels of IL-6, IL-8, and MCP-1 to modulate skin inflammation [61].

Recently, the secretome of MSCs have drawn more attention as a mechanism governing skin repair and regeneration by managing their microenvironment with growth factors and cytokine secretion [21,22]. In wound healing, ADSC extracellular vesicles were involved in the migration and proliferation of DF and keratinocytes, including collagen and elastin deposition [8,11,12]. In the same way, the authors reported similar positive effects of ADSC-conditioned media on skin aging manifestations [38,62,63,64] and similar results were found when using the conditioned media derived from other MSCs [65,66].

This interplay between ADSC secretion and the other epidermal progenitors seem to orchestrate the hierarchical process of regeneration and repair by locally inducing ADSC-residents’ crosstalk in aging or after injury [11,65,67,68]. Nevertheless, when derived from the same microenvironment, location, and was seen in the same human dermo–epidermo skin substitute (DESS), MSCs from forskin, palmer skin, and tonsils presented similar graft morphology, keratinocyte proliferation and differentiation, and maturation [69].

## 4. ADSCs Advancements in Skin Therapy

ADSCs are found in higher frequencies in liposuctions, they can differentiate into multiple cell lineages, and they are safely transplanted in both autologous and allogeneic settings. Used expanded or not-expanded or within the SVF, the performing therapeutic applications were very promising, suggesting that their potential use in regenerative medicine is becoming a real therapeutic alternative for wound management strategies to various skin-related disorders and to rescue the disadvantages or limitations in conventional treatments [70]. ADSCs appeared more proliferative and maintained their potential to differentiate into cells of mesodermal origin during aging, which was contrary to BM-MSCs [71,72].

Skin injury is expected to be the first application where using ADSCs might be justified to different extents. Their use has been shown to promote fat tissue survival after transplantation with fat used in the case of soft tissue augmentation surgery, in breast augmentation, and facial tissue defects [73,74,75,76,77]. Additionally, their multiple applications increased the rate of cell proliferation, thus leading to accelerated wound healing [78]. Their use helped the emergence of a near-natural skin concept in terms of appearance, texture, color, and metabolic properties. Other investigations have used autologous or allogenic ADSCs in the treatment of Crohn’s disease, critical limb ischemia, diabetic foot ulcers, and burns [63,79,80,81,82]. In combination with fat or platelet-rich lysate, skin engraftment, and reconstitution have been improved in human and animal models [83,84,85]. Additionally, platelets lysate appeared to support ADSC proliferation and differentiation in vitro and might replace the use of fetal bovine serum during cell expansion [86].

In wound healing and burns, these cells contribute to the phases of tissue repair and skin reconstitution to different extents, either by their ability to proliferate and differentiate or by secreting different growth factors and cytokines involved in these phases (Figure 1). Tissue engineering has added many insights into the way ADSCs can be used in skin repair. Klar et al. performed a 3-dimension DESS, characterized as the most near-natural skin substitute that is also able to present the patient skin coloration by adding his own melanocytes to pigment and to preserve the skin from UV [70,87,88,89].

Different biomaterials have been investigated to support the expansion and the growth of skin cells with the aim of obtaining an artificial skin substitute, and most of them used SVF or ADSCs. The latter has proven to be efficient in the formation of cell sheets by increasing their ECM secretion in the presence of a collagen scaffold or not [89,90,91].

### 4.1. Inflammation Phase

By their immune effects, ADSCs modulated inflammation during wound healing to initiate tissue reconstitution [71,72,83,84]. The inflammatory status was amplified by aging, where the accumulation of senescent cells resulted in the increase in proinflammatory factors such as IL-6, IL-8, and TNF-α, which are mostly associated with chronic inflammation. The production of IL-β paves the way for this inflammatory-aging [92,93]. ADSCs secrete TGF-β, and together with IL-1β and IL-6, increase macrophage recruitment and their polarization from M1 to M2 [94], followed by a phase of secretion of anti-inflammatory cytokines [79,95]. In vitro, conditioned media of ADSCs have been reported to stimulate macrophages and to increase secretion on TNF and IL-10 anti-inflammatory cytokines, which stimulated wound healing [96].

The presence of a developed vascularization strongly impacted this macrophage’s phenotype changes, likely through the paracrine effects of dermal cells, and principally, the ADSCs. In the early stages of the burn’s rat model, CD68+ nitric oxide synthase+ (iNOS) proinflammatory M1 macrophage infiltrated the DESS graft and were polarized to CD68+ CD206+ M2 macrophages with a less inflammatory profile and contributed to wound healing three weeks later [95]. Vascularization also helped the infiltration of skin graft by monocytes/macrophages CD11b+ CD68+, induced by endothelial cells lining the blood vessels [97]. These endothelial cells interacted with ADSCs to increase their proinflammatory growth factor secretion as IL-6, IL-8, and MCP-1 [61].

### 4.2. Migration

A lot of evidence suggests that ADSCs and DF are able to migrate into injured skin or the upper epidermis to regenerate senescent cells and to repopulate the skin. SDF-1 was revealed to be the constitutively and mostly expressed protein involved in human skin cell migration in normal and in damaged tissues [29,98,99]. This protein appeared to be even more upregulated in some human skin disorders such as psoriasis, basal cell carcinoma, and squamous cell carcinoma. DF is the major source of SDF-1, revealed by double immunostaining and heat shock protein 47 (HSP47), and is considered to be the fibroblast’s markers. ADSCs have proved to overexpress SDF-1 to be recruited into the damaged site [99,100], and inversely, this factor stimulates their paracrine, proliferation, and migration abilities [101,102,103,104]. When treated with the adipose tissue extracellular fraction, fibroblasts improved their motility through increasing CD44 and N-cadherin expression [105]. SDF-1 activated keratinocytes proliferation by interacting with its receptor CXCR-4 [29]. This SDF-1/CXCR-4 axis is recognized to be the key tool in cell homing and migration in normal and injured tissue and provides increased insights into cancer development and metastasis.

### 4.3. Angiogenesis

A lot of evidence suggests the involvement of ADSCs in inducing neoangiogenesis in tissue repair and wound healing. They secrete factors such as VEGF, PDGF, IGF, HGF, b-FGF, SDF-1, TGF-β, and GDF11, proving the efficiency of stimulating the differentiation of ADSCs and DF into endothelial cells, as reported recently [78,79,94,106,107,108]. This differentiation appeared to be associated with apoptosis prevention and found application, particularly in treating critical limb ischemia [79]. Inducing microenvironment changes by the overexpression of SDF-1 in DF led to keratinocyte proliferation and ADSC recruitment, providing that a neovascularization has taken place together with the increase of ECM secretion, thus promoting the remodeling phase of healing [29,105,109]. Adipose tissue extract paved the way for ADSCs and accelerated wound healing and improved angiogenesis via the increased vessel density [110].

Other evidence of the potential differentiation of ADSCs into endothelial progenitor cells has been reported recently. ADSCs interacted with endothelial cells and macrophages to increase MCP-1 and VEGF secretion in order to regulate angiogenesis [61,111]. Indeed, these factors provide for the formation of new vessels during the proliferative phase. Cross interactions between ADSCs, DFs, and macrophages added to the growth factors and ECM secretion derived from the microenvironment to became propitious to accelerating neovascularization. Moreover, ADSCs expressed HIF-1α regulating VEGF gene expression in endothelial cells [112,113]. Others have confirmed the stimulation of ADSCs proliferation and keratinocytes chemoattraction via HIF-1α and VEGF, which are involved in angiogenesis and are facilitated by MMP [114,115]. Recently, SDF-1 has been released, together with HIF-1α in exosomes of ADSCs overexpressing miR-21, and potentially promotes angiogenesis [116]. Similar results were reported recently when using multiple injections of autologous ADSCs to burn wounds in the animal model, where angiogenesis was found to have an enhanced association with the increase in VEGF expression [78].

In chronic radiation wounds, ADSCs also stimulated DF to proliferate and increase VEGF secretion, consequently enhancing the capillary density [117]. This vascularization is crucial because it permits the granulation tissue formation and graft recovery through a spatiotemporal infiltration of the dermal layer with granulocytes (HIS48^+^) and host-derived monocytes/macrophages (CD11b^+^, CD68^+^) [97]. In this work, endothelial progenitors and ADSCs derived from human SVF were used to develop the vasculature in a collagen type I-based dermal component in vitro.

### 4.4. Proliferation/Reepitelialization

During the proliferation phase, the cytokines and chemokines secreted by these ADSCs cells were involved in several fibroblast characteristics, such as cell proliferation, migration, specifically collagen synthesis, and other ECM proteins connected with tissue repair and regeneration [63,118,119]. In addition, administered intravenously with intramuscular or topical applications, ADSCs were found to accelerate reepithelialization of the wounded area in vitro [120]. Indeed, conditioned media from ADSCs, umbilical cord-, and amniotic fluid-MSCs significantly enhanced the proliferation of DF [121]. The involvement of MSCs and DF are essential for the cascade of the factors related to skin regeneration and reflected the importance of sirtuins and SMAD pathways [122,123]. Other significant involvement is that ADSCs differentiate into DF and this differentiation appeared to be improved, thus accelerating granulation tissue formation, leading to re-epithelialization [15,26,38,94]. When cultured in the presence of EGF and high Ca2+ level, ADSCs differentiate into keratinocyte-like cells presenting keratinocyte markers such as involucrin, cytokeratin-5, cytokeratin-10, fillagrin, and stratifin. Additionally, they acquire the capacity to form a stratified structure, as observed in normal skin [124]. The healing improvement has also been reported when using stimulated ADSCs with GDF11 or when using ADSCs-conditioned media containing GDF11 [26,125]. Healing also appeared to be dependent on the number of ADSCs present or administered on burnt skin [78].

Basically, TGF-β has an important role in inducing wound healing, mainly by activating ADSCs to differentiate, to increase their ECM secretion, to regulate melanin production, for the transfer into keratinocytes, as well as the activation of keratinocyte differentiation and maturation. More importantly, the TGF-β implication might be amplified and upregulated by GDF11 to accelerate skin cell production and maturation and cicatrization after injury. Indeed, TGF-β associated with MMP-9, was reported to play a key role in remodeling and wound closure [126,127]. In the same way, GDF11 improved tissue repair by increasing cell proliferation and production, cell migration and angiogenesis, and ECM production, leading to rapid healing in animal models or in vivo [66,105,125]. These promising properties are suggested to be the result of combinatory relationships between these two factors and balancing them is the tool key of regulating tissue repair and regeneration [53].

## 5. ADSCs Issues in Skin Aging

There is evidence suggesting the superiority of ADSCs in improving and increasing dermal thickness and in reducing wrinkles, which is more likely by inducing paracrine DF and angiogenesis [20,26,128]. Administrated intradermally to aged skin, skin texture and wrinkles, as well as dermal thickness, were found to have improved eight weeks after treatment [129]. Additionally, ADSCs from younger donors exhibited higher proliferation capacity when compared to those derived from elderly ones, while maintaining their differentiation capacity within both donors [130].

The TGF-β superfamily growth factors appeared to play a major role in regulating skin mechanisms and preventing their impairment after injury or during aging. TGF-β by targeting ADSCs and DF, is largely reported to be a crucial participant in skin regeneration by inducing skin cell proliferation, migration, enhancement of angiogenesis, regulating melanogenesis, and accelerating wound healing and closure [131]. However, GDF11 largely takes advantage of TGF-β by increasing the similar tissue repair mechanisms, but at a younger cellular status. Another factor, GDF15, is also directly associated to skin photoprotection [132].

TFG-β and GDF11 are both shown to be part of the secretome of ADSCs and their action on skin repair might result from a specific crosstalk between these factors, DF and ADSCs [53].

Recently, FGF has been proposed as a therapeutic option to avoid skin aging aspects and to counter the cellular responses related to aging [133,134]. Recombinant FGF also found application in improving angiogenesis and wound healing by strongly activating fibroblasts and keratinocyte proliferation [135,136]. FGF-2 and b-FGF are other factors interacting with the keratinocyte growth factor (KGF) to reduce wrinkles by targeting fibroblasts and keratinocytes modulating in the same way as the normal process of angiogenesis, tissue regeneration during wound healing, and significantly improving the anti-aging process [2,137,138]. According to these considerations, FGF might interact independently with skin cells, especially DF and keratinocytes, but their actions remain concomitant to TGF-β and GDF11 to improve skin structure and aging aspects [53].

Several reports have confirmed that dermal composition in cells modulate the melanin-producing enzymes by secreting cytokines, especially TGF-β regulation, and thus skin pigmentation and melanocytes maturity [70,139]. Similarly, in in vitro 3D skin constructs, the use of recombinant GDF11 significantly reduced melanin production resulting from GDF11 and TGF-β interactions [125]. These reports led us to the thought that modifications in dermal composition within aging might reflect the impairment of the balance between TGF-β and other factors, or the inefficacy of GDF11 to ensure the proliferation and production of younger and new skin cells.

## 6. ADSCs Issues between Hopes and Limits

The introduction of ADSCs as a key tool in stem cell-based therapies in wound healing appears to be successful and responds to many limitations observed, even with autologous cultured epidermal autografts (CEA), allogenic skin graft, and cellular or acellular skin substitutes. Allogenic fibroblasts were first used to support skin architecture by providing ECM protein secretion, but also by feeding surrounding skin cells by growth factors involved in cell proliferation and tissue regeneration. Using ADSCs within these skin substitutes have permitted the rapid reconstitution, more micro-vasculatures, more elasticity and better dermal cell compositions with high ECM secretion resulting in proliferation, and the remodeling phases of wound healing [52]. Skin has a more satisfying esthetic appearance and function, consequently overcoming the CEA limitations observed, such as recurrent open wounds and fragility, and the extensive ECM protein production leading to scar contractions [91].

BM- or UC-MSCs have been used in some therapeutic advances to achieve complete remission for patients treated for ulcers, scars, and burns [140]. However, the ability of ADSCs have largely proved their superiority for widespread applications in this field. ADSCs took advantage of their MSCs counterparts by their ability to be specifically identified by the CD271 marker [24]. Controlling the rejuvenating ability of ADSCs might be profitable to other tissue and organ transplantations. Using the mechanical and biochemical characteristics of biomaterials, ADSCs can infiltrate the constructed scaffold and differentiate into specialized newborn young cells, thus constituting, completely or partially, the deficient organ unavailable by allogenic donation. The combination of ADSCs and the elastin-like recombinamers (ELR)-based hydrogel has been reported to induce the formation of new blood vessels within the biomaterials, thus providing nutrients and oxygen supports [141].

Recently, miR was reported as a potent regulator of skin cell senescence, cancer, and wound healing. MiR-enriched secretome presented a promising strategy to drive ADSC secretion towards defined therapeutic approaches [54,142,143].

Nevertheless, the point to highly consider is the fact that ADSCs are closely built-in and niched in the ECM and interconnected to other epithelial cells. When collected within liposuctions, their separation from their niche became the first parameter limiting their clinical applications. Indeed, reports have indicated that local anesthetic agents might negatively impact ADSC viability and quantity [144,145,146,147], suggesting that local anesthesia should be avoided. While ADSCs are widely applied in pre-clinical and clinical applications, they might not attain their full efficacy, even when applied in autologous or allogenic settings. This efficacy might be related to the route of administration being mostly realized through intravenous injection [148].

Another point of view is that the ability of the entire administered ADSC to migrate and or to differentiate into skin cells might be challenging. These cells were reported to undergo apoptosis a few days after transplantation [148], making doubtful the implication of patient-associated factors or other epigenetic factors in regulating the fate of ADSCs. Even pathological symptoms and biological markers might appear similar, microenvironment of stem cells remains dependent of their number, their ability to proliferate and differentiate, adding to the fact that their secretion of growth factors is also associated with age and genetic predisposition.

## 7. Summary and Perspectives

Resident ADSCs within the skin are considered to be key regulators and play a crucial roles in tissue repair and regeneration by, on the one hand, replacing, repairing, and regenerating dead or damaged cells, and on the other side, permitting their turn over to provide a continuous recruitment of mature specialized cells from the basal epidermal layer to its outer.

In wound defects, the aims of the strategies altogether converge to ensure patient satisfaction in terms of esthetic appearance and functionality. Based on their great ability to proliferate, differentiate, and migrate adding to their immunomodulatory effect, advancements using skin cellular substitute, biomaterials, and fat graft have been promising in supporting skin repair. However, ensuring complete and functional generation remained difficult to manage with regards to the composition of the repaired dermis and epidermis. Nevertheless, the more enriched secretome the growth factors and cytokines ADSCs are able to secrete makes them more advantageous in participating in skin cell biology and function. These growth factors were reported to autoactivate ADSCs and initiate the whole mechanisms involved in different phases of wound healing [52]. Indeed, ADSCs secretome is composed of proinflammatory and anti-inflammatory growth factors, fibroblasts growth factors including GDF11, TGF-β, b-FGF, VEGF, TLR2, TLR4, IL-10, and MMP involved directly in cell proliferation, and tissue repair and rejuvenation processes [61,111,149,150,151,152]. These factors are consistently responsible for the inflammatory status of the immune skin cells and their impact on the nature and efficacy of healing is more appropriate by modulating the skin microenvironment, also suggesting the implication of ADSCs. TGF-β secreted by utmost skin epithelial cells regulated inflammatory responses via coordinating leukocytes infiltrations [59,153] and through NFkB and SMAD mechanism pathways [154].

Other factors of GDF11, closely involved in increased gene expression, related to ECM production and skin cell proliferation, and also appeared to be a pivotal component of microenvironment regulation. A decrease in IL-1β and other proinflammatory cytokine gene expression was reported [92,125]. The suppression of inflammation severity in psoriasis-associated inflammation occurred by reducing macrophages infiltration in the skin through inhibiting NFkB signaling [155,156] and by abolishing probably TNF-α release [157]. By reducing this NFkB pathway, GDF11 has also a protective effect against apoptosis [155]. Both TGF-β and GDF11 are also specifically implicated in skin inflammatory process during aging or inflamm-aging by down-regulating the gene expression on proinflammatory cytokines [92,93]. This inflamm-aging was principally supported by the accumulation of senescent cells producing proinflammatory cytokines such as IL-6, IL-8, and TNF-α. Moreover, GDF11 was demonstrated to be protective against aging, leading to epidermal turnover and differentiation by triggering SMAD signaling in a TGF-β like fashion.

Targeted cells are the other crucial parameters highlighting the clinical use of ADSCs secretome. Senescent cells secreted more pro-inflammatory cytokines, while the younger ones presented less senescence characteristics, including DNA damage and ROS accumulation, which directly profited in inducing cell rejuvenation and accelerating cell repair and regeneration. Indeed, secretome derived from younger cells is more suitable to increase proliferation compared to the older cell’s one [26], probably due to the decreasing in GDF11 secretion during aging. On the contrary, senescent ADSCs secretome can trigger and reinforce senescence within their microenvironment [27] and this senescent effect is likely transmitted by ligands of TGF-β by mediating changes in the transcriptional program [28]. TGF-β derived ADSCs and skin cells to increase cell repair and regeneration and orchestrate the phases of wound healing. However, while activated with GDF11, ADSCs presented rejuvenating profile and characteristics, leading to a potential use in directly inducing their secretome towards a tissue regeneration or rejuvenation. Nevertheless, the antiaging paracrine effect seemed to be induced to a significant degree, by the combinatorial effects of both GDF11 and TGF-β, which appeared to be regulated in a spatiotemporal pathway. It is probably that both signals vary with age and that the strength of each of them is reciprocal to the sites of the secreted signals and to the length of the exposure to the signal.

Modulation of the inflammatory status of the microenvironment would not be possible without homing and migration of ADSCs and the other immune cells. Finally, controlling the cell enrichment and inflammatory profile of the dermis exhibits a potential advantage in developing strategies in regenerative medicine in both autologous and allogeneic settings [158,159].

Many studies were conducted to identify specific subpopulations of ASCs from different sources and with regard to their immunological profile [148]. The Food and Drug Administration (FDA) also declared that autologous ADSCs from the stromal vascular fraction to be a “drug” due to the use of collagenase during component separation enhancing the need in standardizing the isolation and culture protocols (US Food and Drug Administration (FDA). CFR-code of federal regulations title 21. Part 860: Medical device classification procedures [Internet]. Silver Spring, MD: FDA; 2013.2013 Aug 29).

Based on these considerations, further investigation into the molecular mechanisms regulating ADSC involvement in skin regeneration are needed to increase our understanding on the way the microenvironment can respond to parameters such as inflammation, oxidative stress and cell senescence, and open the way to therapeutic approaches in regenerative medicine.

## Figures and Tables

**Figure 1 ijms-21-01306-f001:**
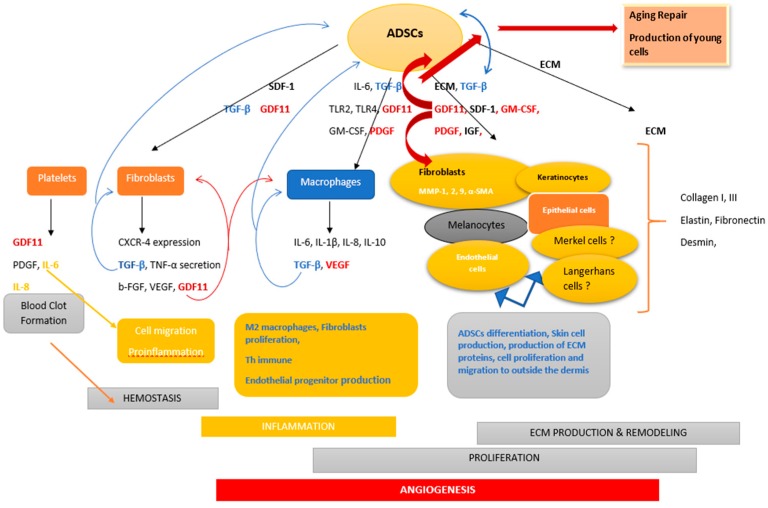
Implication of adipose derived stem cells in the different phases associated with wound healing and in the rejuvenation process. ADSCs act on fibroblasts, macrophages, and skin cells through their secreted growth factors. GDF11 and TGF-β are present in almost all the phases, amplifying fibroblasts, macrophages, and ADSC secretion, leading to immune responses, cell proliferation, and angiogenesis. However, their interactions are more relevant during proliferation phases where GDF11 might induce TFG-β induction in a spatio-temporal status, in addition to boosting fibroblast proliferation, resulting in the production of skin cells presenting young profiles. GDF11: growth differentiation factor, TGF-β: transforming growth factor, ECM: extracellular matrix, PDGF: platelets derived growth factor, Il-1,6,8,10: interleukin-1, TNF-α: tumor necrosis factor-α, b-FGF: basic-fibroblast growth factor, VEGF: vascular endothelial growth factor, CXCR-4: C motif chemokine receptor 4, SDF-1: stromal derived factor-1, TLR2, 4: toll-like receptor2,4, GM-CSF: granulocyte monocyte-colony stimulating factor, IGF: insulin growth factor, MMP-1,-2, -9: matrix metalloproteinase-1, -2, -9, and α-SMA: α-smooth muscle actin.

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
