# Peer review of "Hopes and Limits of Adipose-Derived Stem Cells (ADSCs) and Mesenchymal Stem Cells (MSCs) in Wound Healing"

_ijms, 2020, doi:10.3390/ijms21041306_

Round 1

Reviewer 1 Report

The manuscript entitled "Hopes and limits of adipose-derived stem cells (ADSCs) and mesenchymal stem cells (MSCs) in skin therapy" by Mazini et al., is a review mainly focused on the ADSC-based therapy in skin repair.

In my opinion, the manuscript could be of interest and suitable for publication on IJMS, after a minor revision.

Comments

The title of the review should be more closely related to the content of the manuscript. In fact, the authors focused their attention mainly on the use of ADSCs in skin repair. The term "skin therapy" included in the title appears too generic and suggests skin pathologies in general and not specifically wound healing.

In several parts of the manuscript, an article or a book or a book’s chapter by Mazini et al is cited in (ten times) (Mazini L et al. IntechOpen Editions, in press), but it is not included in the List of References.  So, it is not possible to know the details of this article (Title, authors, publication details). By the way, InTech Open Access Publisher is included in the list of so called predatory journals (see the internet for a list of predatory journals). Thus, according to my opinion, this citation which should be deleted.

Finally, some recent articles/reviews on the topic treated by the authors should be cited, i.e.

Stem Cells Int. 2019 Oct 31;2019:1234263. doi: 10.1155/2019/1234263. eCollection 2019.

Stem Cells Int. 2019 Sep 17;2019:2745640. doi: 10.1155/2019/2745640. eCollection 2019.

Stem Cells Dev. 2019 Nov 1;28(21):1463-1472. doi: 10.1089/scd.2019.0113.

Stem Cell Res Ther. 2019 Jan 31;10(1):47. doi: 10.1186/s13287-019-1152-x.

Stem Cell Res Ther. 2019 Aug 14;10(1):252. doi: 10.1186/s13287-019-1356-0.

Mol Pharm. 2019 Oct 7;16(10):4302-4312. doi: 10.1021/acs.molpharmaceut.9b00639. Epub 2019 Aug 26.

Int J Mol Sci. 2019 Jul 30;20(15). pii: E3721. doi: 10.3390/ijms20153721. Review.

Int J Mol Sci. 2019 Apr 3;20(7). pii: E1671. doi: 10.3390/ijms20071671.

Eur J Pharmacol. 2019 Jan 15;843:307-315. doi: 10.1016/j.ejphar.2018.12.012. Epub 2018 Dec 8. Review.

Author Response

Dar Reveiwer,

Thank you very much for the interest you manifest to my work.

The title of the manuscript has been changed to : “Hopes and limits of adipose-derived stem cells (ADSCs) and mesenchymal stem cells (MSCs) in wound healing” as suggested by the reviewer Thank you very much for advising me on the reality of the IntechOpen Access Publisher. For more transparency, my citation related on this book was deleted. Thank you for the interesting references you suggested. They were added within the text in the underlying locations in the word file with modifications and joined other citations in other places to reinforce the topic.

Reviewer 2 Report

In this review, the authors summarize well characteristics of ADSCs and mechanisms of their actions in the skin. The content is easily understandable and may be useful for readers.

My comments are as follows.

1. In this review, applications of ADSCs for skin disorders are not fully addressed. Besides wound healing and cosmetic applications, clinical usefulness of ADSCs for inflammatory skin diseases, such as atopic dermatitis and psoriasis, may be also interesting for readers.

2. Pages 3, 5 and 9: The authors state that ADSCs induce the secretion of TLR2 and TLR4. Are those TLRs any secretory forms of TLR2 and TLR4?

3. Some grammatical errors should be fixed.

Author Response

Dear Reviewer,

Thank you very much or reviewinf my manuscript. for your comments, please find bellow my replay:

Thank you for your suggestions that we agree completely. The manuscript treated only ADSCs applications for wound healing and cosmetics and in that case we decide to change the title as it was also suggested by another reviewer. The other part of their clinical applications will be investigated in a other paper coming latter. ADSCs have been demonstrated to secrete TLRs factors, we have published a previous work demonstrating that these cells increased their mRNA expression of TLR2 and TLR4 and their relative proteins in the supernatant of expansion culture. Their levels appeared impacted by the duration of the culture and the cryopreservation procedure (Othmani A et al. Cryopreservation Impacts Cell Functionality of Long Term Expanded Adipose-Derived Stem Cells. Journal of Stem Cell Research & Therapy 2019, 9:1 DOI: 10.4172/2157-7633.1000445) Grammatical errors have been corrected.

Round 2

Reviewer 2 Report

The revised manuscript seems to be modified well according to reviewers' comments.